# Role of ^68^Ga-DOTATOC Positron Emission Tomography in Locating Pulmonary Neuroendocrine Tumor Presenting with ACTH-Dependent Cushing’s Syndrome: A Case Report

**DOI:** 10.3390/jcm14248634

**Published:** 2025-12-05

**Authors:** Misako Tanaka, Masakazu Uejima, Kuniaki Ozaki, Maiko Nishigori, Yukako Kurematsu, Kosuke Kaji, Kei Moriya, Tadashi Namisaki, Akira Mitoro, Fumihiko Nishimura, Motoaki Yasukawa, Hitoshi Yoshiji

**Affiliations:** 1Department of Gastroenterology, Nara Medical University, Kashihara 634-8521, Japan; mtanaka00000@yahoo.co.jp (M.T.); kajik@naramed-u.ac.jp (K.K.); moriyak@naramed-u.ac.jp (K.M.); tadashin@naramed-u.ac.jp (T.N.); mitoroak@naramed-u.ac.jp (A.M.); yoshijih@naramed-u.ac.jp (H.Y.); 2Department of Diabetes and Endocrinology, Nara Medical University, Kashihara 634-8521, Japan; ozakunimoonrabit1988@gmail.com (K.O.); maiko.ya.820@gmail.com (M.N.); yukakok@naramed-u.ac.jp (Y.K.); 3Departrment of Neurosurgery, Nara Medical University, Kashihara 634-8521, Japan; fnishi@naramed-u.ac.jp; 4Department of Respiratory Surgery, Hirataka Kohsai Hospital, Hirakata 573-0153, Japan; myasu1007@hotmail.com

**Keywords:** ^68^Ga-DOTATOC-PET, neuroendocrine tumor, ACTH-dependent Cushing’s syndrome

## Abstract

**Background:** In ectopic adrenocorticotropic hormone (ACTH) syndrome, locating the responsible lesion is often challenging. **Case Presentation**: A 68-year-old woman was transferred to Nara Medical University hospital for a detailed investigation of her ACTH-dependent Cushing’s syndrome. Because of hypercortisolism-induced immunosuppression, she subsequently developed severe *Nocardia* pneumonia and was forced to temporarily depend on noninvasive positive pressure ventilation (NIPPV). Intravenous antifungal agents and antibiotics were administered, resulting in significant symptomatic improvement. Metyrapone was administered to suppress excessive cortisol. Contrast-enhanced magnetic resonance imaging of the pituitary revealed a 4 mm sized poorly enhanced area, and microadenoma was suspected. Although cavernous venous sampling was indispensable prior to trans-spheroidal surgery (TSS), this examination could not be performed because of the presence of deep vein thrombosis. TSS was performed for both diagnostic and therapeutic purposes, but hypercortisolism did not improve. Moreover, immunohistochemical findings of the specimen revealed nonfunctional pituitary tumor. **Methods:** We re-evaluated the responsible lesion causing ACTH-dependent Cushing’s syndrome. Fluorine-18 fluorodeoxyglucose (FDG) positron emission tomography (PET) revealed weak and abnormal FDG uptake in the right pericardium, but the possibility of nonspecific uptake could not be ruled out. However, gallium-68 1,4,7,10-tetraazacyclododecane-*N*,*N*′,*N*′′,*N*′′′-tetraacetic-acid-D-Phe^1^-Tyr^3^-octreotide (^68^Ga-DOTATOC)-PET demonstrated the same degree of abnormal uptake; therefore, a functional pulmonary tumor was strongly suspected. **Results:** Video-Assisted Thoracic Surgery (VATS) was performed, and histopathological findings of the specimen revealed a neuroendocrine tumor with positive ACTH staining. After VATS, ACTH and cortisol levels were normalized. **Conclusions**: Here, we report a case of ACTH-dependent Cushing’s syndrome caused by a lung neuroendocrine tumor, in which ^68^Ga-DOTATOC PET was helpful in detecting the functional tumors.

## 1. Introduction

Cushing’s syndrome, particularly ectopic ACTH syndrome, frequently leads to an immunocompromised state and susceptibility to opportunistic infections due to elevated cortisol levels [1,2]. Detecting the location of ectopic ACTH syndrome can be challenging, and obtaining an accurate diagnosis is significantly time-consuming. Gallium-68 somatostatin receptor positron emission tomography (PET) using gallium-68 1,4,7,10-tetraazacyclododecane-*N*,*N*′,*N*′′,*N*′′′-tetraacetic-acid-D-Phe^1^-Tyr^3^-octreotide (^68^Ga-DOTATOC) is a valuable diagnostic tool for patients with neuroendocrine tumors (NETs) [3]. Herein, we present a case of ACTH-producing pulmonary NET with Cushing’s syndrome. In this case, ^68^Ga-DOTATOC PET played a crucial role in identifying the responsible NET lesion, complementing FDG-PET. Yasukawa, one of the co-authors, previously reported this case (in Japanese; only the Abstract is available in English) from a respiratory surgeon’s perspective [4]. Here, we revisit the case from an endocrinological viewpoint.

## 2. Case Presentation

A 68-year-old woman with systemic edema and hypertension was previously admitted to a different hospital, where a blood examination showed hyperglycemia and hypokalemia. Due to excessively high ACTH and cortisol levels, ACTH-dependent Cushing’ syndrome was suspected, and she was transferred to our hospital for a detailed investigation of her condition. Her height and weight were 156 cm and 59 kg (BMI 24.2), respectively. Her blood pressure was 148/87 mmHg, and her pulse was 71 bpm. Physical examination revealed a moon-face, central obesity with thin arms and legs, systemic pigmentation, attenuated skin, and ecchymosis. She had no symptoms of striae, hirsutism, or buffalo hump. The results of the laboratory examination were as follows: serum potassium level: 3.4 mEq/L; blood glucose level: 211 mg/dL; glycated hemoglobin level: 8.6%; ACTH level: 60.9 pg/mL; serum cortisol level: 58.4 μg/dL; and urine cortisol level: 3.5 mg/day (Table 1). A 0.5 mg overnight dexamethasone suppression test revealed ACTH and cortisol levels of 485.7 pg/mL and 114.4 μg/dL, respectively. The 8 mg overnight dexamethasone suppression test revealed ACTH and cortisol levels of 109.3 pg/mL and 569.1 μg/dL, respectively. The Corticotropin-releasing hormone (CRH) stimulation test demonstrated failure of ACTH or cortisol stimulation (Table 2), and the diurnal rhythm of serum cortisol levels was not observed. Computed tomography (CT) of the chest demonstrated an infiltrative shadow in the left upper lobe, suggesting a fungal infection (Figure 1) and a small mediastinal nodule in contact with the right pericardium (initially regarded as a similar fungal lesion). CT of the abdomen and pelvis showed no abnormal lesions, but pituitary magnetic resonance imaging (MRI) revealed a 4 mm sized mass at the left anterior pituitary lobe, suspected to be a pituitary adenoma (Figure 2).

After admission to our hospital, her respiratory condition deteriorated rapidly. Chest CT showed an aggravated infiltrative shadow accompanied by cavity formation (Figure 3), and sputum culture revealed *Nocardia* species and *Aspergillus fumigatus*. The patient was diagnosed with acute exacerbation of fungal pneumonia; therefore, noninvasive positive pressure ventilation (NIPPV) was started, and intravenous antibiotics (imipenem/cilastatin, vancomycim) and antifungal agents (voriconazole) were administered. Systemic steroid pulse therapy was combined simultaneously (Figure 4), and to suppress excessive cortisol, metyrapone was administered, which was started at 750 mg daily and gradually increased up to 2250 mg daily (Figure 5). Because the patient responded well to multidisciplinary therapy, her general condition gradually recovered and NIPPV was withdrawn. Chest CT revealed diminished infiltrated shadow with cavity. Non-suppressed serum cortisol levels after 8 mg dexamethasone administration and non-reactive plasma ACTH under CRH test indicated ectopic ACTH syndrome rather than an ACTH-producing pituitary tumor. On the other hand, an ACTH-producing pituitary tumor could not be completely ruled out on account of the pituitary MRI findings. A cavernous venous sampling test was not performed because of deep vein thrombosis in the lower extremities. After informed consent was obtained from the patient by the endocrinologist and neurosurgeon, we planned to perform transsphenoidal surgery (TSS) to remove the pituitary tumor detected on MRI, regarded as an ACTH-producing tumor. However, the ACTH and cortisol levels did not decrease even a week after TSS (Figure 5). Histopathological findings of the specimen revealed non-tumor pituitary-like tissue, thus indicating that the tumor was not an ACTH-producing pituitary tumor. Therefore, metyrapone was restarted and administered continuously. In due clinical course, ACTH and cortisol levels gradually decreased, suggesting a case of cyclic Cushing’s syndrome. After 4 months of admission, metyrapone was withdrawn because the ACTH and cortisol levels normalized spontaneously (Figure 6). An ectopic ACTH-producing tumor was strongly suspected; therefore, FDG-PET was performed, revealing an 11 mm sized weak uptake at the right pulmonary nodule in contact with the pericardium whose maximum standardized uptake value (SUV max) was 2.4 (Figure 7A). However, nonspecific FDG uptake also appeared to be present in the lung fields, making it difficult to distinguish whether the uptake in the nodule represented a significant lesion or merely an inflammatory lesion. In this case, ^68^Ga-DOTATOC PET was performed to determine if there were other responsible lesions. It showed clearer abnormal uptake (SUV max: 4.1) at a similar lesion where FDG was incorporated, with no clear findings in other areas. Based on the FDG-PET and ^68^Ga-DOTATOC PET results, this pulmonary nodule was regarded as an ACTH-producing tumor (Figure 7B). Wedge resection of the right middle lobe was performed for both curative treatment and definite diagnosis. There were no serious complications, and the postoperative course was uneventful, with ACTH and cortisol levels normalizing after resection (Figure 6). Immunohistochemically, this resected specimen revealed that the tumor was positive for synaptophysin, chromogranin A, CD56, and ACTH; the Ki-67-labeling index of the tumor cells was 2.7% (Figure 8). The final diagnosis was ACTH-producing NET (typical carcinoid) of the lung (pT1N0M0, stage IA), which was considered reasonable based on the imaging findings.

## 3. Discussion

Opportunistic infections and severe bacterial infections are most prevalent in ectopic ACTH syndrome, which is indicated by very high plasma cortisol levels, thereby resulting in high mortality. Infections with Aspergillus species, *Cryptcoccus neoformans*, *Pneumocystis jirovecii*, and *Nocardia asteroides* predominate [5]. Cushing’s syndrome with a very high plasma cortisol level results in a severe immunocompromised state [6], a principal mechanism of this being corticosteroids suppressing inflammation by impeding the access of neutrophils and monocytes to the inflammatory site. Cushing’s syndrome encompasses Cushing’s disease, ectopic ACTH syndrome, and cortisol-producing adrenal adenoma. Of these three, serum cortisol levels are the highest in ectopic ACTH syndrome [2]; therefore, the possibility of this condition should be considered in the case of opportunistic infections due to Cushing’s syndrome. The patient described herein developed severe Nocardia pneumonia and invasive pulmonary aspergillosis within 7 days of admission. During this period, her serum cortisol levels gradually increased; therefore, deterioration of hypercortisolism accelerated immunodeficiency and led to severe pneumonia, which forced us to initiate NIPPV. The clinical, biochemical, and radiographic features of ectopic ACTH-dependent Cushing’s syndrome are often indistinguishable from those of Cushing’s disease [7]. Ectopic ACTH-dependent Cushing’s syndrome tends to have a longer clinical course because NETs, which are the most common etiology, are difficult to detect radiographically on account of their small size and slow progression [8,9]. Based on the available literature, it is estimated that ectopic ACTH-dependent Cushing’s syndrome constitutes 8–18% of all cases of Cushing’s syndrome. Tumors associated with ectopic ACTH-dependent Cushing’s syndrome are most frequently distributed in the lung; pulmonary NET, 21%; small-cell carcinoma, 21% [10]. FDG-PET has been shown to be inferior to CT and MRI in the detection of ectopic ACTH sources because FDG-PET has limited sensitivity for pulmonary NETs due to low uptake of the marker [11,12]. The sensitivity of FDG PET/CT was 66% for low-grade NETs. PET using 11-C-5HTP or ^68^Ga-octreotide may have advantages in this regard. ^68^Ga-DOTATOC is a radiolabeled somatostatin analog for PET/CT that shows specific uptake in tumor cells expressing the somatostatin receptor [13]. DOTATOC has higher somatostatin receptor 2 (SSTR2) affinity [3]. NETs show higher expression of SSTR2 and SSTR5, and octreotide has higher affinity for SSTR2 and SSTR5 [14]. Therefore, tumor imaging using radio-labeled somatostatin analogs such as 111-In octreotide scintigraphy (SRS) and ^68^Ga-DOTATOC PET are useful for detecting NET. SRS has high sensitivity in localizing gastroenteropancreatic NETs, paragangliomas, pituitary tumors, small-cell lung carcinomas, and meningiomas [15]. However, SRS is not always specific for NETs because SRS also detects other SSTR2-expressing tumors and non-neoplastic lesions such as sarcoidosis and rheumatoid arthritis [16]. ^68^Ga-DOTATOC PET is useful for detecting not only NETs but also their metastatic lesions; that is, it is adapted not only for detecting primary lesions but also for deciding on surgical treatment indications [17,18]. ^68^Ga-DOTATOC PET/CT is superior to 111-In-pentetoretide single-photon emission CT for the detection of NET metastases [19,20]. Isidori et al. [21] conducted a systematic review regarding the accuracy of imaging studies in ectopic Cushing’s syndrome. This review mentions that ^68^Ga-SSTR PET has the highest sensitivity (81.8%), followed by CT (66.2%) and MRI (51.5%). In overt cases, CT showed the greatest sensitivity (98.3%), followed by MRI (92.9%), FDG-PET (71.1%), and ^68^Ga-SSTR PET (70%). In contrast, in covert cases, ^68^Ga-SSTR PET showed the greatest sensitivity (100%), followed by FDG-PET (59.4%), MRI (44.8%), and CT (43.6%). These data suggest that CT has the highest sensitivity on initial imaging for localizing the source of ACTH, and ^68^Ga-SSTR PET has the highest sensitivity on subsequent follow-ups [21]. In our study, the ACTH-producing pulmonary NET could not be fully detected via CT and FDG-PET but was accurately detected via ^68^Ga-DOTATOC PET. Ectopic Cushing’s syndrome is difficult to diagnose when it is not immediately overt and the tumor size is small. However, nuclear medicine techniques, such as ^68^Ga-DOTATOC PET, may help in identifying the source of ectopic ACTH production.

## 4. Conclusions

We herein report a case of an ACTH-producing pulmonary NET presenting as Cushing’s syndrome complicated by the presence of severe Nocardia pneumonia. Hypercortisolism due to Cushing’s syndrome sometimes leads to critical opportunistic infections; therefore, localizing the source of ACTH secretion is crucial. However, there are cases in which the primary tumor site remains unknown for a long time. ^68^Ga-DOTATOC PET is useful for detecting NETs; here, it played a key role in strengthening the accurate diagnosis of the primary tumor detected via FDG-PET. This study suggests the usefulness of ^68^Ga-DOTATOC PET for the immediate detection of NETs presenting as covert Cushing’s syndrome.

## Figures and Tables

**Figure 1 jcm-14-08634-f001:**
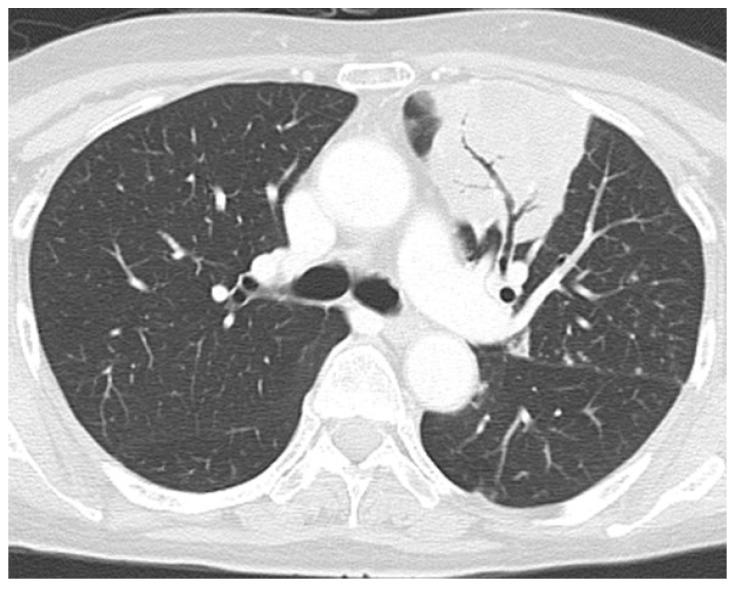
Computed tomography (CT) of the chest demonstrating infiltrative shadow at the left upper lobe, suggesting fungal infection.

**Figure 2 jcm-14-08634-f002:**
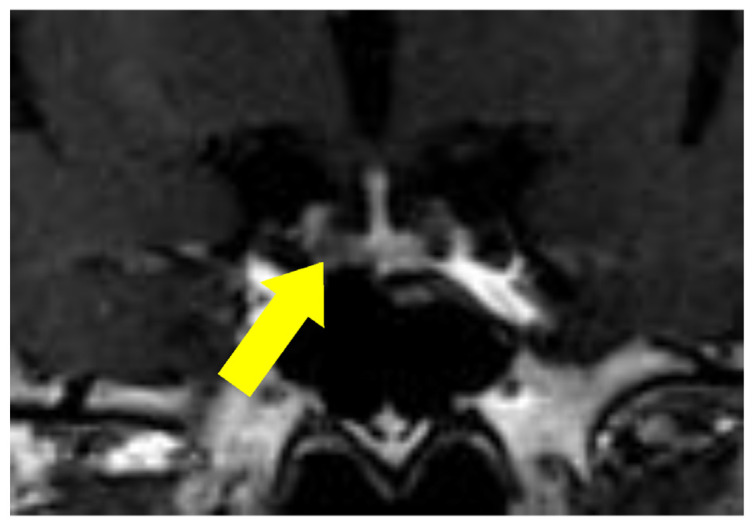
T1 gadolinium-enhanced magnetic resonance imaging (MRI) of the pituitary, revealing a 4 mm sized mass at the left anterior pituitary lobe (yellow arrow), suggesting pituitary adenoma.

**Figure 3 jcm-14-08634-f003:**
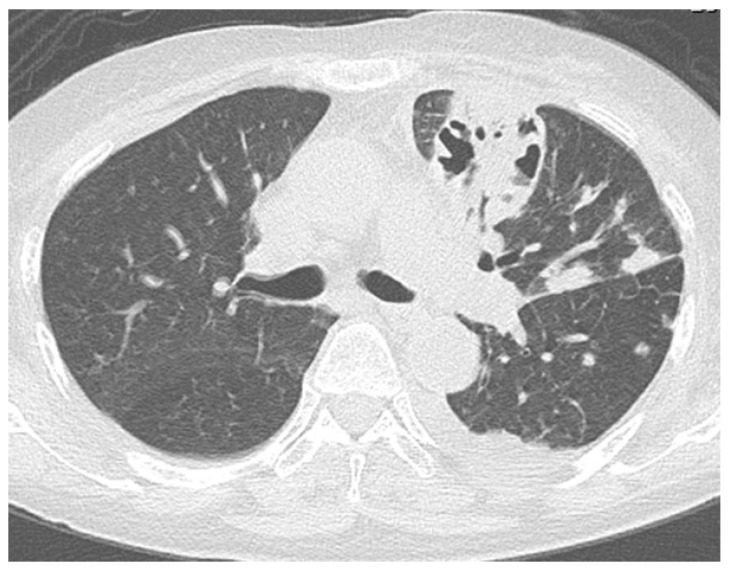
Chest computed tomography (CT) after hospitalization showing aggravated infiltrative shadow accompanied by cavity formation, diagnosed as acute exacerbation of fungal pneumonia.

**Figure 4 jcm-14-08634-f004:**
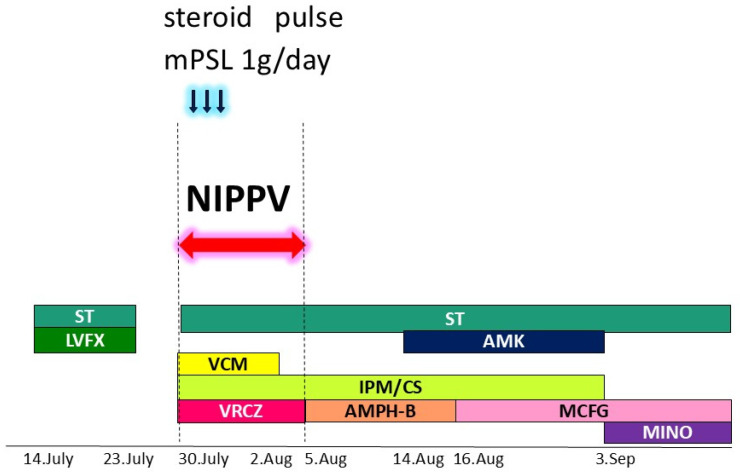
Multidisciplinary therapy was administered, including artificial respiration, antibiotics, antifungal agents, and systemic steroid pulse therapy. Various colors were used to clearly illustrate the types of antibiotics and their duration of use. Blue arrows indicated steroid pulse therapy using 1 gram per day of methylprednisolone. Red arrow and dashed line indicated the period during which NIPPV was performed. NIPPV: noninvasive positive pressure ventilation; ST: Sulfamethoxazole/Trimethoprim; LVFX: Levofloxacin; AMK: amikacin; VCM: Vancomycin; IPM/CS: Imipenem/Cilastatin; VRCZ: Voriconazole; AMPH-B: Amphotericin B; MCFG: Micafungin; MINO: minomycin; mPSL: methylprednisolone.

**Figure 5 jcm-14-08634-f005:**
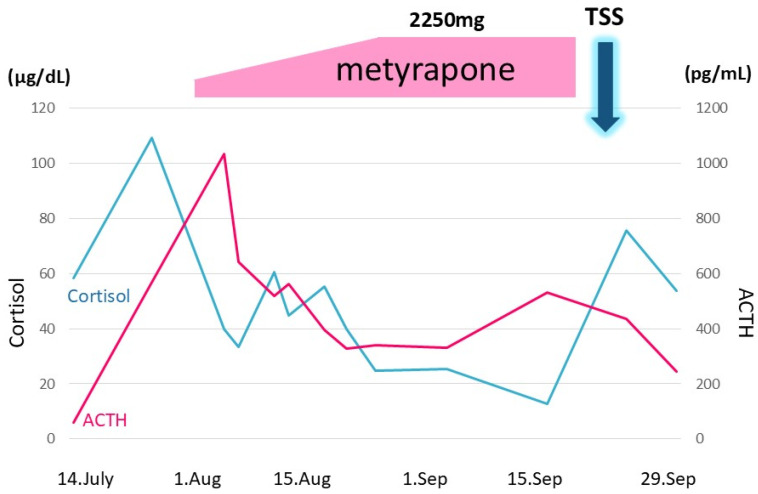
Metyrapone was administered to suppress excessive cortisol. The daily dose was started at 750 mg and gradually increased up to 2250 mg. Transsphenoidal surgery (TSS) was performed on 24 September; however, ACTH and cortisol levels remained unchanged. The vertical axis denotes ACTH and cortisol levels. The horizontal axis denotes time. 7/14: 14 July. The black and gray line graphs indicate serum cortisol and plasma ACTH levels, respectively.

**Figure 6 jcm-14-08634-f006:**
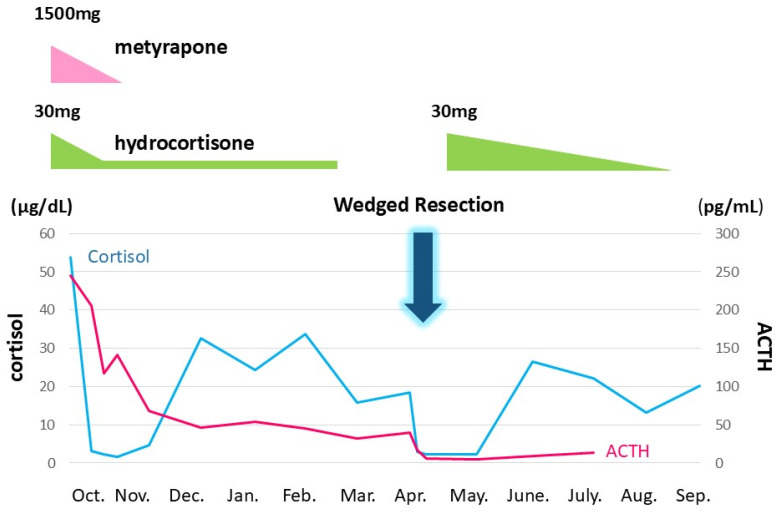
Ectopic ACTH syndrome was controlled in combination with metyrapone and hydrocortisone. In due clinical course, ACTH and cortisol levels gradually decreased, suggesting cyclic Cushing’s syndrome. Right middle lobectomy was performed on 10 April. The vertical axis denotes ACTH and cortisol levels. The horizontal axis is for time. 10/6: 6 October. The black and gray line graphs indicate serum cortisol and plasma ACTH levels, respectively.

**Figure 7 jcm-14-08634-f007:**
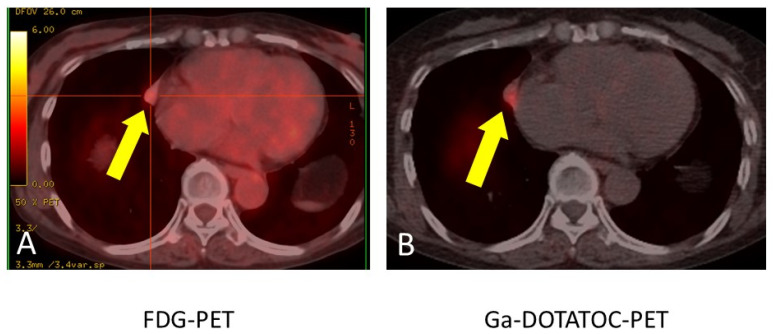
FDG-PET revealed 11 mm sized weak uptake (SUV max: 2.4) at the right pulmonary nodule (yellow allow) in contact with pericardium. The red crosshair was for calculating SUV max (**A**). Gallium-68 DOTATOC PET revealed abnormal uptake (SUV max: 4.1) at a similar lesion (yellow allow) where FDG was incorporated (**B**). This strongly suggested that this nodule was the lesion responsible for ectopic ACTH syndrome.

**Figure 8 jcm-14-08634-f008:**
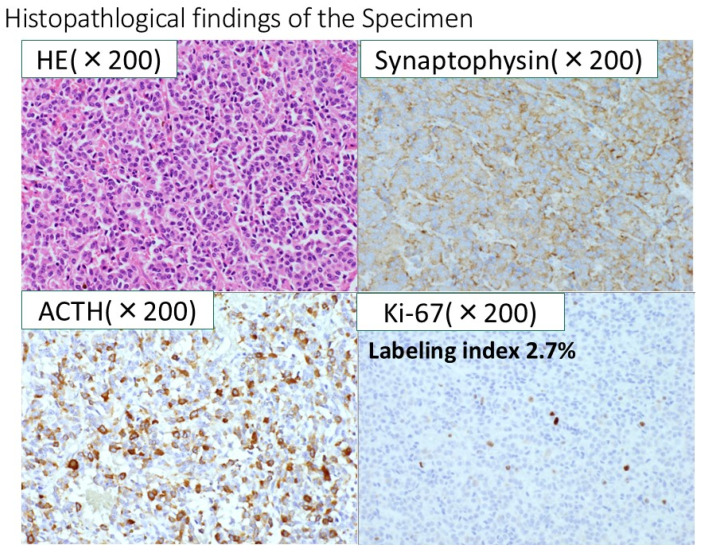
Immunohistochemically, the resected specimen revealed that the tumor was positive for synaptophysin, chromogranin A, CD56, and ACTH. The final diagnosis was ACTH-producing neuroendocrine tumor. The Ki-67-labeling index of the tumor cells was 2.7%.

**Table 1 jcm-14-08634-t001:** Laboratory examination on admission. CBC: complete blood count.

CBC			Biochemistry					
WBC	124 × 10^2^	/μL	CRP	1.0	mg/dL	Na	138	mEq/L
Nuet	118 × 10^2^	/μL	TP	4.6	g/dL	K	3.4	mEq/L
Lym	2.0 × 10^2^	/μL	Alb	2.6	g/dL	Cl	97	mEq/L
Eosi	0.0 × 10^2^	/μL	AST	25	IU/L	Ca	7.8	mg/dL
Neut%	95.2	%	ALT	54	IU/L	Pi	1.9	mg/dL
Lym%	1.7	%	ALP	204	IU/L	T-ch	161	mg/dL
Eosin%	0.0	%	γGTP	28	IU/L	HDL-ch	44	mg/dL
RBC	371 × 10^4^	/μL	LDH	336	IU/L	LDL-ch	92	mg/dL
Hb	10.9	g/dL	UA	2.2	mg/dL	Glucose	211	mg/dL
PLT	189 × 10^3^	/μL	BUN	12	mg/dL	HbA1c	8.6	%
			CRE	0.63	mg/dL	Glycoalbumin	18.8	%
			T-bil	0.6	mg/dL	β-D-glucan	56.1	pg/mL

**Table 2 jcm-14-08634-t002:** Endocrinological examination during admission.

Pituitary Gland				
ACTH		60.9		pg/mL
TSH		0.74		μIU/mL
PRL		16.7		ng/mL
LH		<0.10		mIU/mL
FSH		1.01		mIU/mL
Adrenal gland				
Plasma Renin Activity		0.4		ng/mL/hr
Aldosterone		95.1		pg/mL
Cortisol		58.4		μg/dL
Urine cortisol		3500		μg/day
Daily variation				
	7:00 AM	11:00 AM	5:00 PM	10:00 PM
ACTH (pg/mL)	179.7	213.6	129.4	143.4
Cortisol (μg/dL)	122.4	133.1	87.9	106.2
Dexamethasone (DEX) suppression test				
	DEX 0.5 mg		DEX 8 mg	
ACTH (pg/mL)	485.7		569.1	
Cortisol (μg/dL)	114.4		109.3 (baseline 114.4)	
CRH stimulation test				
	baseline	30 min	60 min	90 min
ACTH (pg/mL)	342.9	267.7	54.9	62.5
Cortisol (μg/dL)	118.2	108.7	110.6	112.9

## Data Availability

Any data relevant to this case that are not presented in this manuscript can be obtained from the corresponding author upon reasonable request.

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
