# Peer review of "Role of 68Ga-DOTATOC Positron Emission Tomography in Locating Pulmonary Neuroendocrine Tumor Presenting with ACTH-Dependent Cushing’s Syndrome: A Case Report"

_jcm, 2025, doi:10.3390/jcm14248634_

Round 1
Reviewer 1 Report
Comments and Suggestions for Authors
The manuscript, titled "The Role of 68Ga-DOTATOC Positron Emission Tomography in the Definitive Diagnosis of Pulmonary Neuroendocrine Tumors Presenting with ACTH-Dependent Cushing's Syndrome," explores the application of 68Ga-DOTATOC positron emission tomography in the definitive diagnosis of pulmonary neuroendocrine tumors that present with ACTH-dependent Cushing's syndrome. A case report" is devoted to a clinical case of ACTH-dependent Cushing's syndrome caused by a neuroendocrine tumor of the lung. This work is of significant interest, as cases of hypercorticism without clinically visible foci that produce adrenal cortex hormones or ACTH are challenging to diagnose. In such cases, the use of gallium as a marker drug is of paramount importance. This communication is of significant importance and will be of interest to clinicians, general practitioners, and individuals involved in the field of endocrinology who are engaged in medical practice.
The manuscript demonstrates a commendable level of structural clarity; however, there are minor inaccuracies in the English language that must be rectified. It is imperative to refine the text's stylistic elements to enhance its clarity and precision. It would be beneficial to receive a description of the immunohistochemistry of this neoplasm, should the authors have the capability to provide such information. It is my conviction that the merits of this work justify its scientific value.
The manuscript, titled "The Role of 68Ga-DOTATOC Positron Emission Tomography in the Definitive Diagnosis of Pulmonary Neuroendocrine Tumors Presenting with ACTH-Dependent Cushing's Syndrome," explores the application of 68Ga-DOTATOC positron emission tomography in the definitive diagnosis of pulmonary neuroendocrine tumors that present with ACTH-dependent Cushing's syndrome. A case report" is devoted to a clinical case of ACTH-dependent Cushing's syndrome caused by a neuroendocrine tumor of the lung. This work is of significant interest, as cases of hypercorticism without clinically visible foci that produce adrenal cortex hormones or ACTH are challenging to diagnose. In such cases, the use of gallium as a marker drug is of paramount importance. This communication is of significant importance and will be of interest to clinicians, general practitioners, and individuals involved in the field of endocrinology who are engaged in medical practice.
The manuscript demonstrates a commendable level of structural clarity; however, there are minor inaccuracies in the English language that must be rectified. It is imperative to refine the text's stylistic elements to enhance its clarity and precision. It would be beneficial to receive a description of the immunohistochemistry of this neoplasm, should the authors have the capability to provide such information. It is my conviction that the merits of this work justify its scientific value.
Author Response
It is imperative to refine the text's stylistic elements to enhance its clarity and precision.
Thank you for pointing this out. From now on, we are going to use English editing service to improve the quality of English.
It would be beneficial to receive a description of the immunohistochemistry of this neoplasm, should the authors have the capability to provide such information. It is my conviction that the merits of this work justify its scientific value.
I appreciate your kind peer review. Immunohistochemistry of the resected specimen revealed positive staining for synaptophysin, chromogranin A, CD56, and ACTH. Therefore, the diagnosis of ACTH-producing neuroendocrine tumor had been made. We added the comment of immunohistochemistry in this manuscript.
We have revised the manuscript based on the review comments from Reviewer 1 and Reviewer 2. Added sections are indicated in red, and deleted sections are crossed out with double lines.

Reviewer 2 Report
Comments and Suggestions for Authors
This manuscript describes a case of ACTH-dependent Cushing’s syndrome secondary to a pulmonary neuroendocrine tumor, in which 68Ga-DOTATOC PET played a relevant role in identifying the ectopic source of ACTH production. The case is clearly presented, and the diagnostic reasoning is generally sound.
However, while this is certainly an example of good clinical practice and multidisciplinary management in a challenging endocrine condition, the overall novelty and originality of the case are limited. Ectopic ACTH production from neuroendocrine tumors is a well-known entity, and the diagnostic contribution of ^68Ga-DOTA-peptide PET imaging has already been reported in several previous publications.
A further concern relates to the authors’ claim that ^68Ga-DOTATOC PET was decisive (“helpful in detecting functional tumors”). However, this statement is not adequately supported by data. No quantitative information is provided (e.g., SUVmax, tumor-to-background ratio), and the qualitative comparison between FDG PET and 68Ga-DOTATOC PET suggests a similar degree of uptake. Without objective evidence that the DOTA-peptide PET provided incremental diagnostic value, the conclusion appears overstated.
Specific comments:
-
Quantitative PET parameters (such as SUVmax) should be included to support the imaging findings.
-
The authors should better clarify in what way 68Ga-DOTATOC PET changed clinical management, since FDG PET had already shown an area of abnormal uptake in the same region.
-
Minor linguistic and formatting revisions would improve the manuscript’s readability.
Author Response
I appreciate your kind peer review and thank you very much for your valuable feedback. We have revised the manuscript based on the review comments from Reviewer 1 and Reviewer 2. Added sections are indicated in red, and deleted sections are crossed out with double lines.
Quantitative PET parameters (such as SUVmax) should be included to support the imaging findings.
Thank you for your suggestion. We reanalyzed the data and found that SUVmax of the pulmonary NET was 2.4 in FDG-PET, and 4.1 in DOTATOC-PET. These data were added to our main text and legend.
Without objective evidence that the DOTA-peptide PET provided incremental diagnostic value, the conclusion appears overstated.
The authors should better clarify in what way 68Ga-DOTATOC PET changed clinical management, since FDG PET had already shown an area of abnormal uptake in the same region.
In response to this comment, we have revised the main text. We have changed our assertion from “DOTATOC-PET, not FDG-PET, was useful for the definitive diagnosis of NETs” to “The combination of FDG-PET and DOTATOC-PET enabled accurate diagnosis.”
Minor linguistic and formatting revisions would improve the manuscript’s readability.
Thank you for your advice. From now on, we are going to use English editing service to improve the quality of English.

Round 2
Reviewer 2 Report
Comments and Suggestions for Authors
The authors have appropriately revised the manuscript. Although its originality is limited, the multidisciplinary approach and the appropriate use of different tracers (FDG/⁶⁸Ga-DOTA) make this clinical case interesting from an educational and illustrative standpoint